# Mechanisms of Metastatic Tumor Dormancy and Implications for Cancer Therapy

**DOI:** 10.3390/ijms20246158

**Published:** 2019-12-06

**Authors:** Christiana M. Neophytou, Theodora-Christina Kyriakou, Panagiotis Papageorgis

**Affiliations:** 1European University Research Centre, 1516 Nicosia, Cyprus; C.Neophytou@research.euc.ac.cy; 2Department of Life Science, European University Cyprus, 1516 Nicosia, Cyprus; t.kyriakou@euc.ac.cy

**Keywords:** dormancy, organ-specific metastasis, tumor microenvironment, extracellular matrix, stromal cells

## Abstract

Metastasis, a multistep process during which tumor cells disseminate to secondary organs, represents the main cause of death for cancer patients. Metastatic dormancy is a late stage during cancer progression, following extravasation of cells at a secondary site, where the metastatic cells stop proliferating but survive in a quiescent state. When the microenvironmental conditions are favorable, they re-initiate proliferation and colonize, sometimes years after treatment of the primary tumor. This phenomenon represents a major clinical obstacle in cancer patient care. In this review, we describe the current knowledge regarding the genetic or epigenetic mechanisms that are activated by cancer cells that either sustain tumor dormancy or promote escape from this inactive state. In addition, we focus on the role of the microenvironment with emphasis on the effects of extracellular matrix proteins and in factors implicated in regulating dormancy during colonization to the lungs, brain, and bone. Finally, we describe the opportunities and efforts being made for the development of novel therapeutic strategies to combat metastatic cancer, by targeting the dormancy stage.

## 1. Introduction

Cancer is the second leading cause of death worldwide [1]. Despite the development of new therapeutic approaches and significant improvement in survival rates in the last twenty years, metastatic disease, primarily to the bone, lungs, and brain, remains incurable and is the main cause of death for most cancer patients [2]. 

Cancer progression leading to metastatic spread of cancer cells is a highly complex, yet poorly understood process and consists of a series of discrete events which initiate as primary tumor cells that lose epithelial characteristics and acquire mesenchymal-like features, a process also known as epithelial to mesenchymal transition (EMT), degrade the basement membrane and invade the interstitial matrix [3]. Subsequently, cancer cells intravasate in the circulation through lymphatic or hematogenous routes and survive via secretion of growth factors, cytokines, and protective interactions with platelets [4]. Upon arresting in the narrow capillaries of the target organ, cancer cells can disrupt the endothelial junctions, extravasate into the surrounding tissue and remain initially dormant until conditions facilitate metastatic colonization [5]. While much progress has been made to highlight some of the specific processes which are critical for the initiation of the metastatic spread, the molecular mechanisms involved in the fatal late stages remain far from fully elucidated. 

During the multistep metastatic cascade, only a fraction of cancer cells, termed disseminating tumor cells (DTCs), acquire the necessary genetic and epigenetic alterations to complete transition to the subsequent stage. Despite this significant attrition, a small percentage of cancer cells can eventually reach secondary organs during the evolution of the disease. However, upon extravasation, disseminated cells often remain quiescent in distal organs undergoing long periods of latency, also known as the dormancy period [6]. The term dormancy refers to two conditions: (1) single cells or small clusters of DTCs that are in a quiescent state and (2) microscopic lesions or micrometastases that do not grow in size due to similar rates of proliferation and apoptosis of their constituent cancer cells [7]. 

Cancer cell outgrowth at secondary sites is a highly inefficient process and delays the development of macrometastases [3]. During this time, cancer cells are not easily detectable. In addition, since they are in a non-proliferative state, they are inefficiently targeted by chemotherapeutic drugs. These cells may start proliferating and eventually lead to the development of metastatic disease even years following surgical removal of the primary tumor and treatment [8,9,10,11]. It is noteworthy that disease recurrence happens after almost a decade of being “cancer-free” in many patients while, for example, more than 67% of breast cancer deaths occur beyond the 5-year survival window [12]. Clinical evidence demonstrates that DTCs detected in patients prior to the development of macrometastases may contain fewer genetic aberrations compared to primary or metastatic tumors, suggesting that dissemination of tumor cells is often an early event during cancer progression [13]. This notion is supported by experimental evidence indicating that Her2-driven breast cancer cells may metastasize and remain dormant at a secondary site even prior to detectable growth of the primary tumor [14]. In addition, studies using pancreatic cancer mouse models suggested that EMT and cancer cell dissemination to the liver may occur earlier than tumor formation in response to inflammation [15]. These observations underlie the need to decipher the complex mechanisms that govern metastatic cancer cell dormancy and the escape from this latent state, whereas at the same time highlight the importance to exploit this window of opportunity for effective treatment of cancer metastasis. 

## 2. Mechanisms Regulating Metastatic Dormancy

Disseminated tumor cells (DTCs) that have spread to a secondary site can enter a state of dormancy by exiting the proliferative cycle or by attaining a balanced state of proliferation and apoptosis. During this dormancy period, metastatic cells can accumulate genetic and/or epigenetic aberrations that allow them to optimally adapt to the host microenvironment (Table 1). Once the conditions are favorable, cells emerge from dormancy and initiate colonization [16]. 

### 2.1. Mechanisms That Sustain Metastatic Dormancy

Slow cycling or arrested DTCs activate self-imposed dormancy programs that allow them to adapt to the new microenvironment at the metastatic site and remain unaffected by therapies targeting proliferating cells. F-box/WD repeat-containing protein 7 (FBXW7) is a protein encoded by the *FBXW7* gene that functions as a substrate recognition component of a Skp1-Cul1-F box-type (SCF-type) E3 ubiquitin ligase. FBXW7 restrains the cell cycle through the ubiquitylation and proteasomal degradation of cell cycle promoters, including cyclin E and c-Myc. It is highly expressed in various types of stem cells and promotes dormancy by inhibiting cell cycle entry in vivo [17]. Recently, a role for FBXW7 in maintaining breast cancer dormancy has been uncovered. The ablation of FBXW7 in breast cancer cells using mouse xenograft and allograft models caused DTCs to exit their quiescent state and to start proliferating. Importantly, the ablation of FBXW7 and subsequent re-initiation of cell cycle progression, rendered cancer cells sensitive to paclitaxel, suggesting that a combined therapeutic approach involving genetic targeting of FBXW7 with chemotherapy could be a promising approach [18]. Moreover, leukemia inhibitory factor receptor (LIFR), promotes dormancy of disseminated breast tumor cells in the bone. LIFR acts by activating signal transducer and activator of transcription 3 (STAT3) and suppressor of cytokine signaling (SOCS). Loss of the LIFR or STAT3 enables otherwise dormant breast cancer cells to downregulate genes associated with dormancy, quiescence and cancer stemness, reactivate proliferation, and colonize to the bone [19].

Autophagy, a physiological mechanism often activated following metabolic stress under nutrient deprivation conditions, leads to the degradation of the cytosol, organelles, and misfolded proteins to establish proper energy balance as well as to recycle macromolecules and dysfunctional organelles. Recently, this process has also been implicated in the survival of dormant cancer cells since inhibition of autophagy in these cells may eliminate them to prevent recurrence of breast cancer [20]. More specifically, autophagy-related 7 (ATG7) has been identified to be essential for activation of autophagy in vivo; knockdown of ATG7 was shown to decrease metastatic burden while autophagy blockade specifically targeted dormant breast cancer cells leading them towards apoptotic cell death [20]. 

Mitogen-activated protein kinase (MAPK) kinase 4/c-Jun NH2-terminal kinase (JNK)-activating kinase (MKK4/JNKK1/SEK1), referred to as MKK4, was initially characterized as a metastasis suppressor in prostate and ovarian cancers [21,22]. MKK4 has been shown to activate p38 through its kinase activity and suppress the metastasis of ovarian cancer cells in vivo [23]. Metastatic cells undergoing dormancy have been found to exhibit elevated p38 activity [24]. It is therefore likely that MKK4 may contribute to maintaining dormancy by regulating p38. 

Activation of the canonical nuclear factor kappa-light-chain enhancer of activated B cells (NF-κB) pathway may also be implicated in the promotion of the dormant phenotype in breast cancer cells expressing estrogen receptors (ERs) [25]. A constitutively active form of the inhibitor of NFκB (IκB) kinase β (CA-IKKβ), inhibited estradiol-dependent cell proliferation in vitro and tumor growth in vivo, while co-activation of both ER and IKKβ promoted migration and invasion in vitro and metastasis in vivo [25]. Downregulation of the C-X-C motif chemokine receptor 4 (CXCR4) in breast cancer cells metastasized to the lung, has also been associated with sustaining the dormant phenotype [26]. 

Paired-related homeobox transcription factor (PRRX1) has been associated with the activation of EMT program and in maintaining the dormancy phenotype in head and neck squamous cell carcinoma (HNSCC) patients [27]. PRRX1 was found to be upregulated in invasive primary tumors of HNSCC patients and to promote EMT by activating Transforming growth factor-β1 (TGF-β1) signaling; PRRX1 was found to sustain dormancy in HNSCC cells in vivo by downregulating the expression of miR-642-3p which is associated with tumorigenesis and cell growth. PRRX1 overexpression diminishes miR-642-3p levels which mediates dormancy via transforming growth factor-β2 (TGF-β2) and p38 [27]. 

Finally, the kisspeptin 1 (KISS1) metastasis suppressor protein has been found to strongly inhibit pulmonary and intraperitoneal metastasis in xenograft models of various cancer types, including melanoma, breast and ovarian cancer [28,29,30,31]. Kisspeptin, the protein product of the *KISS1* gene which binds to the G protein-coupled receptor, GPR54, has been proposed to mediate its anti-metastatic effects via this interaction. However, other findings suggest that the effects of KISS1, or its mimetics, could be mediated through a different yet unidentified receptor, and that KISS1 secretion is required for maintaining DTCs in a dormant state [32].

### 2.2. Mechanisms That Promote Escape from Dormancy

The escape from dormancy is often directed by intrinsic changes in expression patterns of the disseminated cancer cells but may also be dependent on certain characteristics and interactions with the host tissue. In the liver, hepatic stellate cells stimulate emergence of dormant disseminated breast cancer cells by secreting soluble factors, including interleukin-8 (IL-8) and monocyte chemoattractant protein-1 (MCP-1). These factors have been shown to promote the proliferation of breast cancer cells even under serum-starvation conditions possibly via extracellular-signal-regulated kinase (ERK) pathway activation [33]. 

In human breast cancer cells, specific markers have been identified that support the ability of DTCs to bypass senescence and reinitiate growth. (TGF-β) has been implicated in promoting metastasis and in regulating the dormant state of cancer cells. The expression of Inhibitor of differentiation (Id) family of proteins (ID) is controlled by TGF-β. These are transcription factors implicated in promoting proliferation and migration as well as inhibiting cell differentiation in many types of cancer [34]. The expression of ID1 and ID3 in cell clusters of basal or triple-negative subtype breast tumors is required to reactivate and sustain proliferation of cancer cells in the lungs during metastatic colonization [35,36]. Coco, a secreted antagonist of TGF-β ligands, has also been shown to facilitate the escape of dormancy in disseminated cells in the lungs by blocking lung-derived paracrine bone morphogenetic protein (BMP) signaling. Coco induces metastatic relapse of breast cancer cells to the lung but not to the bone or brain in patients suggesting that metastasis initiating cells need to overcome organ-specific signals in order to escape dormancy [37]. Furthermore, prostate cancer DTCs to the bone marrow manage to escape their dormant state and initiate proliferation by downregulating TGF-β2 expression and by activating its downstream target myosin light chain (MLC) via MLC kinase (MLCK) [38]. 

### 2.3. Epigenetic Alterations That Regulate Metastatic Dormancy

Due to the fact that cancer cell dormancy is a reversible state, it has been suggested that epigenetic changes may also play critical roles in controlling the switch from quiescence to proliferation. Interestingly, epigenetic modifications have been found to not only reactivate cells but also to keep them in a suppressed steady state. 

Epigenetic regulation of selected genes has been identified as a mechanism promoting the growth of micrometastatic lesions. In estrogen receptor-positive (ER^+^) breast cancer, mitogen- and stress-activated protein kinase-1 (MSK1) appears to play an important role in metastatic dormancy. Low levels of MSKI are associated with early metastasis; however, at the secondary site, MSK1 maintains cancer cells in a steady state by promoting luminal differentiation. Acting downstream of stress-activated kinase p38, MSK1 controls the expression of the GATA binding protein 3 (GATA3) and forkhead box A1 (FOXA1)transcription factors through epigenetic regulation of their chromatin status in respective promoter regions [39]. Earlier work revealed the role of p38 in establishing dormancy via epigenetic mechanisms in epidermoid carcinoma, by regulating the expression of 46 genes including 16 transcription factors. Activation of p38 induced the expression of p53 and BHLHB3, whereas it inhibited c-Jun and FoxM1 expression; these transcriptional changes correlated with cell quiescence [40]. 

Furthermore, epigenetic upregulation of the orphan nuclear receptor N2RF1 is shown to promote dormancy of disseminated head and neck as well as prostate cancer cells, via induction of the pluripotency gene *NANOG* in the bone marrow [41]. 

## 3. Microenvironment

Interactions of DTCs with the microenvironment at the secondary sites upon extravasation, determine whether or when metastatic cells will colonize. Signals received from the extracellular matrix (ECM) either keep cells in an inactive state or drive their progression into macrometastases (Table 1). Some tissue microenvironments are more hospitable than others for cancer cells; these sites, such as the lungs, bones and brain, are organs where metastatic lesions are more frequently detected in patients. We describe below the currently known mechanisms activated in different sites that keep metastatic cells in a dormant state or promote their proliferation. 

### 3.1. The Role of the Extracellular Matrix Components in Regulating Dormancy

Even though DTCs are able to disseminate and survive at secondary sites, they could remain inactive or dormant due to regulatory pathways activated by local microenvironments.ECM proteins play a key role in sustaining DTCs in a dormant state or allowing tumor growth. By deciphering the mechanisms that maintain these cells growth-suppressed or allow for their outgrowth, we may develop more effective therapeutic approaches to keep DTCs in a perpetual dormant state or activate apoptotic signals to eliminate them.

Early studies showed that ECM protein fibronectin (FN) can determine whether DTCs can remain in a dormant state by interacting with the urokinase plasminogen activator receptor (uPAR) in cancer cells. uPA/uPAR proteins physically associate with alpha5beta1 (α5β1) integrin and increase adhesion of cells to fibronectin. However, low levels of uPAR in human carcinoma HEp3 cells, were found to decrease adhesion and promoted the dormancy phenotype [42]. Furthermore, it has been shown that high levels of uPAR activate the integrin alpha5beta1 which generates persistently high levels of active ERK. ERK is a mitogenic extracellular regulated kinase necessary for tumor growth in vivo. ERK is also activated by FN fibrils which suppress the p38MAPK pathway. In contrast, when uPAR levels are low and FN fibrils are absent, cells enter into a dormant state [43]. The regulation of dormancy by FN based on the ERK/p38 ratio, which is controlled via the uPAR/integrin alpha5beta1 axis, was shown in vivo using cell lines of different cancer types, including breast, prostate, melanoma, and fibrosarcoma [24]. When grown in 3-D culture conditions, cells with dormant behavior transitioned from quiescence to proliferation which was dependent on the phosphorylation of myosin light chain by MLC kinase (MLCK) by integrin β1. The activation of MLCK leads to cytoskeletal reorganization with f-actin stress fiber formation while its inhibition significantly reduces metastatic burden in vivo [44]. 

Deposition of type I collagen (Col-I) in the microenvironment of the secondary site has also been shown to facilitate the escape of cancer cells from dormancy through induction of fibrosis. Col-I induces activation of SRC and focal adhesion kinase (FAK) via β1-integrin. This leads to actin stress fiber formation and facilitates the growth of quiescent mouse mammary cells [45]. Col-I crosslinking, catalyzed by the extracellular amine oxidase Lysyl oxidase (LOX), may enhance metastatic outgrowth. LOX’s primary function is to catalyze the covalent crosslinking of fibers in the extracellular matrix, creating a stable microenvironment and supporting metastatic growth. Increased LOX activity, accompanied by fibrillary collagen crosslinking, has been shown to contribute to metastatic seeding in vivo [46]. Moreover, therapeutic targeting and downregulation of LOX led to a 50% decrease in pulmonary metastatic burden and increased survival of mice [46,47]. 

### 3.2. Lung Microenvironment Controls the Metastatic Cell Dormant State 

Lungs are some of the most frequently affected target organs for metastatic disease. Common cancers that metastasize to the lungs include breast, colon, prostate, bladder cancer, neuroblastoma and sarcoma. Cancer cells migrating to the lungs are often driven out of quiescence and into metastatic outgrowth via interactions with the surrounding microenvironment. 

Conditions in the lung microenvironment such as hypoxia and inflammation are shown to favor the development of macrometastases. Lysyl oxidase Like 2 (LOXL2), a member of the LOX family, can facilitate ECM remodeling and induce EMT [48,49]. In addition, LOXL2 expression in breast cancer cells has been associated with a more aggressive phenotype and enhanced metastatic potential [50]. Expression of LOXL2 in MCF-7 breast cancer cells colonizing to the lung induced their transition from dormancy to metastatic outgrowth. The expression of LOXL2 can be driven by the hypoxic conditions in the tumor microenvironment [51]. In addition, inflammation in the lung microenvironment may lead to Tank-binding kinase-1 (TBK1)-dependent promotion of proliferation of dormant breast cancer cells. TBK1, a downstream effector of the miR-200c-driven pathway, facilitates EMT and invasiveness of lung cancer cells by controlling synthase kinase-3β (GSK-3β) phosphorylation and zinc finger E-box-binding homeobox 1 (ZEB1) expression [52,53].

Periostin (POSTN) is a component of the extracellular matrix expressed by fibroblasts in normal tissues. It has been shown that POSTN expression is induced in the lung by infiltrating tumor cells and this allows them to initiate colonization. POSTN recruits Wingless (Wnt) ligands thereby increasing Wnt signaling in cancer cells migrating to the lungs. Primary lung fibroblasts upregulated POSTN in response to tumor-derived transforming growth factor-β3 (TGF-β3) and TGF-β2. POSTN interacted with Wnt1 and Wnt3A, which boosted Wnt signaling and facilitated cancer cell growth [54]. In a head and neck squamous cell carcinoma (HNSCC) model, TGF-β2 activated MAPK p38α/β, therefore reducing the ERK/p38 signaling ratio. This induced DEC2 and transforming growth factor-β receptor-I (TGFβR-I)-dependent quiescence that involved downregulation of cyclin-dependent kinase 4 (CDK4) and activation of p27 [55]. In the lungs, a metastasis-permissive microenvironment with low TGF-β2 levels awakened DTCs from dormancy followed by metastatic growth [55]. 

Vascular cell adhesion molecule 1 (VCAM-1) has been shown to interact with macrophages expressing α4 integrins to induce Akt (protein kinase B) expression and lung metastasis [56]. Specifically, breast cancer cells that infiltrate leukocyte-rich microenvironments, such as the lungs, have a survival advantage when they overexpress VCAM-1. VCAM-1 anchors metastasis-associated macrophages via α4-integrins to cancer cells, triggering Akt activation, thereby protecting cancer cells from proapoptotic cytokines, including TNF-Related Apoptosis Inducing Ligand TRAIL [56]. 

### 3.3. Brain Metastasis: Microenvironmental Factors Regulating Dormancy

Circulating cancer cells infiltrating the brain microvasculature initially encounter endothelial cells. In addition to acting as the initial barrier during cancer cell brain invasion, endothelial cells and their basement membrane may also support the growth of brain metastases. In addition, brain-resident cells such as microglia and astrocytes, provide a unique environment with paracrine growth factors that affect the development of metastatic brain tumors.

Breast cancer cells have been found to extravasate to the brain exclusively from capillaries. During brain infiltration, cancer cells are trapped in narrow vessels and the extravasation process often requires several days. “Arrested” cancer cells within the capillaries were shown to induce diverse astrocytic and microglial responses which caused local changes of the tumor microenvironment that promoted or restricted their progress into macrometastases [57]. More specifically, breast cancer cells escaping dormancy in the brain were found to overexpress the metalloproteinase-9 (MMP-9) protein which is known to have pro-angiogenic and growth-promoting functions in brain tumors [58]. In addition, MMP-9, secreted by activated astrocytes, was increased in the immediate vicinity of extravasating cancer cells. Astrocytes were also found to have elevated expression of Glial fibrillary acidic protein (GFAP) and nestin, two proteins which are thought to help to maintain the shape of cells and provide a supporting microenvironment. This evidence suggests that activated astrocytes surrounding the tumor cells may enable the development and growth of brain metastatic lesions [57].

Brain metastasis also occurs frequently in melanoma patients and represents a significant cause of death from this disease [59]. Brain micrometastases comprising dormant melanoma cells have been found to harbor specific transcriptional signatures. The differential expression of 35 genes as described by Izraely et al., including the levels of cysteine-rich protein 61 (CYR61) and of preferentially expressed antigen in melanoma (PRAME), are suggested to control the ability of disseminated melanoma cells to respond to microenvironment-derived signals in the brain [60]. 

Extrinsic dormancy also includes “angiogenic dormancy”, referring to the concept that cancer cells remain quiescent and undetectable for years before an “angiogenic switch” reactivates them and allows for further tumor growth. Pro- and anti-angiogenic factors produced by the tumor and its microenvironment determine whether the tumor will vascularize and continue to grow [61]. The mechanisms of angiogenic dormancy in metastatic cancer have not been fully characterized. Angiomotin, a protein that mediates the antiangiogenic activity of the endogenous angiogenesis inhibitor angiostatin, is elevated in dormant tumor cells and maintains dormancy of metastatic cells [62,63,64]. During brain metastasis formation, melanoma as well as lung carcinoma cells may find optimal survival conditions at their perivascular position. The awakening and growth of these dormant cells requires high levels of vascular endothelial growth factor A (VEGF-A), suggesting an important role for angiogenesis in the early stages of brain metastasis [65]. This evidence is consistent with the proposed concept that the angiogenic switch is required for the transition of cancer cells from a dormant to an actively proliferating phenotype [66,67].

### 3.4. Metastasis to the Bone: The Role of the Bone Marrow and the Bone Microenvironment in Dormancy

The bone marrow and bone microenvironment, acting as a translational area for DTCs as well as a secondary metastatic site, have been extensively studied for their ability to regulate dormancy.

#### 3.4.1. Bone Marrow and the Regulation of Dormancy

During metastasis to the bone, DTCs infiltrate the highly vascularized bone marrow (BM) where they may remain in a latency phase which can last for years prior to entering a more aggressive active phase. This lag period is regulated and modified by numerous factors in the microenvironment of the bone marrow stroma [68,69]. Several cell types within the BM microenvironment have been shown to promote dormancy. Osteoblasts within the BM secrete niche factors that promote the quiescence of hematopoietic stem cells (HSCs). DTCs displace resident HSCs in the BM and may also become quiescent by binding to these niche factors. For example, osteoblasts secrete growth arrest–specific 6 (GAS6) which can bind to the AXL receptor expressed by prostate cancer cells. In addition, prostate DTCs bind Annexin II, a protein expressed by both osteoblasts and endothelial cells that is involved in diverse cellular processes, including mediating HSC adhesion within the niche [70]. This binding results in upregulation of the AXL family of receptors (Axl, Mer, and Tyro3) and of TGF-β receptors that promote dormancy [71,72]. AXL is also implicated in production of TGF-β2, which enhances GAS6 expression and promotes DTC dormancy [73]. In a recent study, *AXL* was identified as a key gene highly expressed in dormant myeloma cells localized in specific niches of bones; when targeted by small molecule inhibitors, downregulation of AXL promoted cell proliferation and escape from dormancy [74]. When breast cancer cells are localized in the microvasculature of the BM, they are facilitated by endothelial cells that produce thrombospondin-1 (TSP-1), a tumor suppressing ant-angiogenic factor that induces quiescence [75]. Loss of TSP-1 expression at neovascular tips not only allows but accelerates micrometastatic outgrowth. TGF-β1 and POSTN have been identified as tumor-promoting factors derived from endothelial tip cells [75].

Osteoblasts also secrete CXC-chemokine ligand 12 (CXCL12), also known as stromal cell-derived factor 1 (SDF-1). This binds to CXCR4 G-protein coupled receptor which is expressed both on HSCs and DTCs and promotes cellular adhesion in the bone marrow [76,77]. Recently, it has been suggested that during breast cancer colonization to the bone CXCL5 is released from bone marrow cells to activate its receptor CXCR2 expressed on the surface of cancer cells to enable their proliferation [78]. Dormant breast cancer cells have also been found to occupy distinct vascular regions within the BM that are rich in E-selectin. Breast cancer cell interaction with E-selectin was shown to be critical for their mobility and entry into the BM [79]. Bone morphogenetic protein 7 (BMP-7) secreted from BM stromal cells can inhibit proliferation of prostate cancer stem-like cells by activating p38 MAPK signaling and increases expression of N-myc downstream regulated gene 1 (*NDRG1*), a metastasis suppressor gene [80]. 

Macrophages within the ΒΜ stroma may also affect the development of dormancy in DTCs. Specifically, M2-like macrophages that are characterized by tumor promoting, immunosuppressive properties, can form gap junctional intercellular communication (GJIC) with cancer cells reducing their proliferation, causing cycling quiescence, and resistance to carboplatin. In contrast, exosomes derived from macrophages with tumor suppressive M1 phenotype, activate NF-κB to reverse dormancy in quiescent breast cancer cells [81]. 

Extracellular vesicles derived from mesenchymal stem cells (MSCs) residing in the bone marrow may also promote metastatic cell dormancy. Specifically, the MSCs may interact with breast cancer cells via extracellular vesicles that contain microRNAs, conveying a negative influence on their proliferation and increased adhesion [82]. Cancer cells originating from the breast can form typical osteolytic metastases in the BM by secreting parathyroid hormone-related protein (PTHRP), tumor necrosis factor-α (TNFα), interleukin 6 (IL-6) and/or interleukin 11(IL-11). PTHRP, IL-6, IL-11, and TNFα stimulate osteoblasts to release receptor activator of nuclear factor-κB ligand (RANKL) which induces the formation of osteoclasts [83,84,85,86]. 

#### 3.4.2. The Role of the Bone Microenvironment in Dormancy

Osteoblastic niche within the bone microenvironment can promote prostate cancer dormancy via Wnt/β-catenin signaling. Co-culture of prostate cancer cells with osteoblasts or with conditioned media from osteoblasts inhibited their proliferation. Osteoblast-induced tumor cell dormancy was shown to be mediated via Wnt5a that is present in high levels in the microenvironment. This involved activation of non-canonical receptor tyrosine kinase like orphan receptor 2/ Siah E3 ubiquitin protein ligase 2 (ROR2/SIAH2) signaling that suppresses canonical Wnt/β-catenin signaling [87]. 

Within the endosteal niche, metastasized myeloma cells are kept in a dormant state by interacting with bone-lining cells or osteoblasts, and subsequently may become activated to form colonies by engaging with osteoclasts. The dormant population of cells was shown to undergo a limited number of cell divisions due to the downregulation of genes that control cell cycle progression [88].

Differentiated osteoblasts located around the tumor cells in the bone induced dormancy in prostate cancer cells by secreting proteins including growth differentiation factor 10 (GDF10) and TGFβ2 [89]. TGFβ2 and GDF10 can promote cell cycle arrest by binding to the TGF-βRIII receptor expressed by prostate cancer cells. This binding induces transforming growth factor-β receptor-III (TGFβRIII) to activate phospho-p38MAPK, which phosphorylates retinoblastoma (Rb) protein and blocks prostate cancer cell proliferation by upregulating p27. The importance of the TGFβRIII-p38MAPK-Rb signaling axis in maintaining dormancy was evaluated in patients with prostate cancer; lower TGFβRIII expression correlated with increased metastatic potential and decreased survival rates [89]. 

Breast cancer metastasis to the bone may be promoted by NF-κB-mediated aberrant expression of vascular cell adhesion molecule 1 (VCAM-1). VCAM-1 has been implicated in the formation of macrometastasis of breast cancer cells. The mechanism of action involves interaction with the cognate receptor integrin α4β1 and recruitment of monocytic osteoclast progenitors to elevate local osteoclast activity within micrometastases [90]. 

## 4. Therapeutic Approaches against Dormancy

Since the duration of metastatic dormancy offers a unique window for therapeutic intervention, understanding the mechanisms underlying dormancy, either promoting or escaping the quiescent state in DTCs, may allow us to develop therapies that will limit disease recurrence [7]. One approach would be to block the communication of cancer cells with stromal cells of the microenvironment. For example, by disrupting CXCL12/ CXCR4 binding, DTCs are mobilized from the bone marrow, initiate cell cycle progression and become more susceptible to combined treatment with standard chemo- or nanotherapeutics (Table 2). One study showed that oncolytic viruses expressing the CXCR4 antagonist inhibit breast cancer metastases in a mouse model [92]. Another approach could involve maintaining DTCs in a perpetual dormant state. Administration of agents that promote dormancy, such as thrombospondin 1 (TSP1), a glycoprotein secreted from vascular endothelial cells, which has been shown to reduce the proliferation of invasive ductal carcinoma (IDC) cells [93], may prevent the eventual metastatic outgrowth. Angiostatin, a naturally occurring angiogenesis inhibitor currently undergoing clinical trials as a potential anti-cancer agent, was initially found to induce and sustain dormancy of primary tumors in mice [94]. Since SRC has been found to promote escape from dormancy, SRCinhibition by Saracatinib (AZD0530; AstraZeneca), an orally active, dual SRCfamily kinase–AB1 (SFKABL) inhibitor, was able to prevent the Col1-induced proliferation of dormant tumor cells and induced the upregulation and nuclear localization of the cell cycle inhibitor p27. The combination of Saracatinib with an ERK1/2 inhibitor was able to induce apoptotic cell death [95]. 

The specific targeting of dormant cancer cells that reside in the bone marrow represents another challenging approach as in this region the endogenous hematopoietic stem cells also reside. As previously mentioned, cancer cells can interact with mesenchymal stem cells (MSC) within the bone marrow, priming them into releasing exosomes with distinct miRNA content which facilitates their quiescence. One nontoxic therapeutic strategy tested in an immunodeficient mouse model of dormant breast cancer, indicated that systemic administration of MSC loaded with antagomiR-222/223 sensitized breast cancer cells to carboplatin-based therapy and increased host survival [96]. 

Targeting of the lysophosphatidic acid receptor 1 (*LPA1*) gene that has been implicated in promoting metastasis, via the specific inhibitor Debio-0719, was shown to induce dormancy at the secondary tumor site in a breast cancer model [97]. Epigenetic modifiers have also been studied for their therapeutic potential by sustaining dormancy. Combination of 5-azadeoxycytidine (5-Aza-C), a DNA demethylating agent, and retinoic acid was found to upregulate the master receptor NR2F1 and subsequently induce the expression of pluripotency genes *SOX9, RARβ*, and *NANOG* that led to quiescence in head and neck squamous cell carcinoma (HNSCC) cells [41].

Manipulation of the expression of specific genes in dormant cancer cells has been found to alter their responsiveness to treatment. Cancer stem cells (CSC), referring to the tumor-initiating population of cells within a tumor that share common features with stem or progenitor cells, may remain in a quiescent, dormant state and contribute to disease recurrence and chemoresistance. Manipulation of the levels of Fra-1, a member of the Fos transcription factor family, may assist in driving cells out of dormancy and rendering the tumor sensitive to treatment. Knockdown of Fra-1 led to increased chemoresistance in CSC tumors and a concurrent increase in tumor size; overexpression of Fra-1 resulted in decreased tumor incidence and in increased chemosensitivity [98]. Therefore, Fra-1 may be useful in predicting patient response to treatment. In addition, p38-induced cancer cell quiescence in epidermoid carcinoma HEp3 cells can be inhibited by RNAi-mediated downregulation of BHLHB3 or p53 [40]. 

The role of the immune system in regulating dormancy of cancer cells in primary and metastatic sites has been highlighted by many studies and is often referred to as “immunologic dormancy” [99]. Adaptive immunity is responsible for maintaining tumor cells in a state of functional dormancy. Specifically, IL-12, IFN-γ, CD4^+^, and CD8^+^ T cells have been implicated in retaining cells in a dormant state; depletion of these components in mice treated with the carcinogen 3′-methylcholanthrene, led to progressively growing sarcomas [91]. In a mouse model of melanoma, dormancy in the lung was associated with reduced proliferation of DTCs mediated partly by cytostatic CD8^+^ T cells. Depletion of CD8^+^ T cells significantly accelerated outgrowth of visceral metastases [100]. These findings suggest that immunostimulatory therapies may be effective in preventing subsequent development of metastasis by prolonging DTC dormancy. 

The recent discovery and FDA approval of immune checkpoint inhibitors has marked a new era of hope for cancer therapy. However, the development of immunotherapy against dormant disease is challenging, as quiescent DTCs have developed mechanisms to evade the immune system. Dormant DTCs downregulate MHC class I, which is essential for CD8^+^ T cell recognition [101]; in addition, their microenvironment may suppress the immune system by protecting cancer cells from oxidative stress or through the expression of checkpoint ligands such as PD-L1 and secretion of immunosuppressive cytokines such as IL-6 [102,103]. However, therapies that mobilize the immune system against dormant DTCs are being developed. Vaccines targeting tumor-associated blood vessel Ags (TBVA) activate T-cell dependent immunity capable of either inducing tumor regression or extending overall survival by sustaining dormancy [104]. Vaccination of leukemia-bearing mice with cells transduced with CXC-chemokine ligand 10 (CXCL10) induced natural killer (NK) cells to express programmed death-ligand 1(PD-L1), thereby activating T cells that eliminated dormant cells [105]. Combination of 5-Aza-C with tumor-sensitized T cells and CD25^+^ NKT (Natural Killer T) cells was also effective in targeting metastatic dormant mammary cancer cells. 5-Aza-C induced the expression of highly immunogenic cancer testis antigens in the tumor and reduced the frequency of myeloid-derived suppressor cells (MDSCs). The presence of CD25^+^ NKT cells rendered T cells resistant to remaining myeloid-derived suppressor cells [106]. This combinatorial approach significantly prolonged survival of animals bearing metastatic tumor cells. Antibodies against VCAM-1 and integrin α4 were also shown to inhibit breast cancer metastasis to the bone [90].

Modulation of the immune system through dietary changes may also reduce cancer risk and metastatic burden. Low weight has been correlated with decreased cancer incidence. A prevalent hypothesis is that cancer progression is promoted by obesity-associated inflammation [107]. Obesity caused infiltration of neutrophils in murine lungs that led to increased breast cancer metastasis to this site. Granulocyte-macrophage colony-stimulating factor (GM-CSF) and interleukin 5 (IL5) were responsible for pro-metastatic effect of obesity. However, this effect was reversible as mice placed on a low-fat diet had reduced lung inflammation and metastasis [108]. Finally, release of systemic inflammatory signals upon surgical removal of primary tumors has been suggested to stimulate metastatic colonization. Consistent with this evidence, perioperative administration of nonsteroidal anti-inflammatory drugs (NSAIDs) in mouse models, significantly inhibited the outgrowth of dormant metastatic breast cancer cells [109]. 

There are several ongoing clinical trials targeting the population of persistent DTCs in hopes of eradicating dormant cells. In one study, patients with localized breast cancer that have received anthracycline-containing chemotherapy undergo aspiration in the BM 8 to 10 months after the last chemotherapy cycle. If DTCs are detected, patients undergo treatment with Docetaxel to reduce the risk of persistent DTCs (NCT00248703, Phase II). Results so far have been encouraging, with DTC eradication in 79% of patients and enhanced metastasis-free survival. In breast cancer patients that have undergone standard of care treatment, the CLEVER pilot trial (NCT03032406, Phase II), currently recruiting, is investigating a combination treatment of hydroxychloroquine and everolimus to target persistent DTCs. Denosumab, a human antibody specific to receptor activator of nuclear factor kappa-B ligand (RANKL), is being investigated in several clinical trials for the prevention and treatment of bone metastases. In women with early stage breast cancer that have completed cytotoxic chemotherapy, investigators hypothesize that treatment with denosumab will decrease the number of DTCs, prevent cancer cell migration, and promote cancer cell death by changing the bone into a “hostile” environment (NCT01545648, phase II). In another active and recruiting study, 5-Aza-C, which has been found to induce dormancy in certain cancers, and alltrans retinoic acid (ATRA), are being tested in combination in prostate cancer patients with recurrent disease based on rising PSA only. The primary objective is to evaluate the progression-free rate at the end of a 12-week treatment (NCT03572387, Phase II). Finally, an ongoing trial is looking to evaluate the effects of trastuzumab therapy against HER2 expressing DTCs in the BM (NCT01779050, Phase II).

## 5. Future Perspectives

Currently, the presence of dormant disease is validated via detection of disseminating tumor cells using bone marrow aspiration. The ability to locate dormant cancer cells is of utmost importance to determine disease recurrence as well as to increase metastasis-free survival. The detection of circulating tumor cells in the blood is a non-invasive approach to evaluate disease burden. Different methodologies for the detection of CTCs have been shown to have prognostic relevance in patients with metastatic disease [112]. Even though malignant cells located at secondary sites following extravasation will differ in their expression pattern compared to cancer cells in the circulation system, information derived by analyzing the genetic background of CTCs may predict disease outcome and may be valuable in determining optimal patient treatment. Expression of insulin-like growth factor-1 receptor (IGF1R) in CTCs of patients with breast cancer is associated with favorable outcomes in the early disease stage, suggesting that IGF1R expression is correlated with lower metastatic potential [113]. Evaluation of proliferation and apoptosis markers in CTCs of women with breast cancer revealed that the former are elevated on relapse while the latter are increased during clinical dormancy. In addition, in patients who remain disease free, apoptotic CTCs are more frequently encountered during follow-up compared to those with subsequent late relapse. These results suggest that measuring the levels of proliferation and apoptosis markers in CTCs during clinical dormancy is important for determining the risk for detectable disease relapse [114]. Circulating microRNAs, including microRNA (miR)-21, miR-23b, miR-190, miR-200b, and miR-200c, evaluated in the plasma by RT-qPCR, have differential expression among relapsed and non-relapsed breast cancer patients. Studies suggest that during clinical dormancy, miRNAs represent potential circulating biomarkers that may predict detectable disease recurrence [115]. 

## 6. Conclusions

Mechanisms controlling metastatic dormancy involve regulation of genes expressed by DTCs themselves, including genetic and/or epigenetic means of control, as well as mechanisms regulated by the tumor microenvironment. Several underlying pathways have been proposed to either sustain dormancy, making these DTCs more difficult to detect and eliminate, or facilitate the escape of cells from this dormant state at distal sites and formation of macrometastases. These mechanisms often appear to be organ-specific, suggesting that depending on the metastatic site, such as the lung, bone, or brain, dormancy is regulated by distinct components of the respective host tissue microenvironment (Figure 1). Most importantly, the prolonged duration of this stage as evident from clinical and experimental observations, strongly suggests that metastatic cell dormancy should be viewed as a valuable window of opportunity for therapeutic intervention. Therefore, thorough elucidation of the mechanisms that control cancer cell dormancy, either promoting or escaping the quiescent state in DTCs, may lead to the development of innovative therapies that will limit disease recurrence.

Targeted therapeutic approaches including small molecule inhibitors, siRNA-mediated gene silencing, epigenetic modulators, or immunotherapy have shown encouraging pre-clinical efficacy to either keep the cells in a perpetual dormant state or enable them to reenter proliferation so that they become sensitive to concomitant administration of chemotherapeutic or other cytotoxic drugs. Future research should concentrate on unraveling the distinct mechanisms governing the development and maintenance of organ-specific metastatic cell dormancy that will pave the way for the development and clinical testing of novel therapies that do not elicit or minimize off-target effects. 

## Figures and Tables

**Figure 1 ijms-20-06158-f001:**
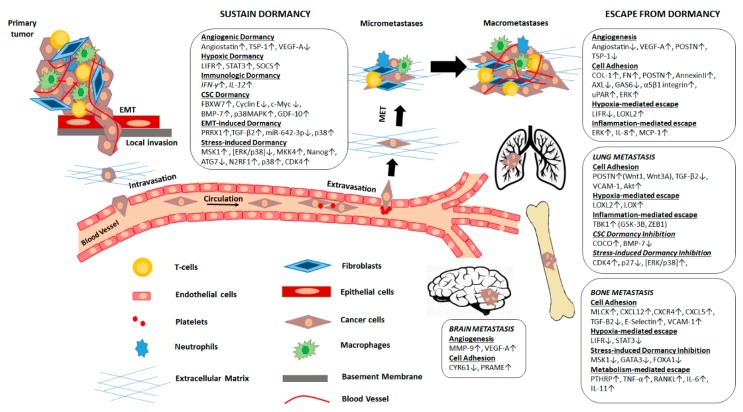
Key factors and related processes that regulate organ-specific metastatic cell dormancy and reactivation. Dissemination of cancer cells from the primary site by activation of EMT program and invasion through the ECM into the circulation is often an early event during tumor progression. A fraction of CTCs survive in systemic circulation and become trapped in narrow capillaries until extravasation occurs in distal sites. Once DTCs reach a target organ site they enter a quiescent, dormant state, via activation of specific transcriptional programs, for periods which may last for even years. Permissive interactions and signals exchanged between cancer and stromal cells within the unique conditions of the host tissue microenvironment may eventually favor the reactivation of cell proliferation and angiogenesis to promote metastatic outgrowth and expansion. Escaping from dormancy is defined by organ-specific secretion of growth factors, cytokines, and ECM components that orchestrate a dynamic interplay between DTCs, immune cells, fibroblasts, and endothelial cells which allow the development of clinically relevant macrometastatic lesions. EMT: epithelial to mesenchymal transition, ECM: extracellular matrix, CTCs: circulating tumor cells, DTCs: disseminated tumor cells.

**Table 1 ijms-20-06158-t001:** Factors (gene, protein, cytokine, growth factor) implicated in the mechanisms sustaining dormancy or escape from dormancy.

Factor	Mechanism	Regulation	Cancer Type	Metastatic Site	Model	Ref
**Mechanisms that Sustain Dormancy**						
Fbxw7	Cell cycle control	Increased levels	Breast	Lung	Mouse	[18]
LIFR	Hypoxia	Increased levels	Breast	Bone marrow	Mouse/Human	[19]
ATG7	Autophagy	Increased levels	Breast	Lung	Mouse/Human	[20]
MKK4	Apoptosis, proliferation, differentiation	Increased levels	Ovarian	Intraperitoneal sites	Mouse	[23]
IKKβ	Inflammation	Overactivation	Breast	Multiple sites	Mouse	[25]
CXCR4	Cell cycle control, Inflammation, Cell survival	Decreased levels	Breast	Lung	Mouse	[26]
PRRX1	EMT	Increased levels	HNSCC	Lymph nodes	Mouse/Human	[27]
KISS 1	Hormone regulation	Increased levels	Melanoma Breast Ovarian	Lung Intraperitoneal sites	Mouse	[28,29,30]
MSK1	Differentiation	Increased levels	Breast	Bone	Human	[39]
N2RF1/NANOG	Development Differentiation	Increased levels	HNSCC Prostate	Bone marrow	Human	[41]
TGF-β2	Development Morphogenesis	Increased levels	HNSCC	Bone marrow	Mouse	[55]
GAS6/AXL	Apoptosis Differentiation	Increased levels	Prostate	Liver, Lymph node, Bone	Mouse/Human	[71]
BMP-7	Morphogenesis Differentiation	Increased levels	Prostate	Bone	Mouse/Human	[80]
Wnt5a	Development	Increased levels	Prostate	Bone	Mouse/Rat/Human	[87]
GDF10/TGF-β2/ TGF-βRIII	Cell cycle regulation	Increased levels	Prostate	Bone	Mouse/Human	[89]
IFN-γ, IL-12	Immune response	Increased levels	Sarcoma	Multiple sites	Mouse	[91]
Mechanisms that Promote Escape from Dormancy						
IL8/MCP-1	Inflammation	Increased levels	Breast	Liver	Ex vivo	[33]
ID1/ID3	Proliferation Differentiation	Increased levels	Breast	Lung	Mouse	[35]
Coco	Morphogenesis	Increased levels	Breast	Lung	Mouse	[37]
MLCK	Proliferation, Actin stress fiber formation	Constitutive activation	Prostate Breast	Bone marrow Lung	Human Mouse	[38,44]
Col-I	Induction of fibrosis	Increased levels	Breast	Lung	Mouse	[45]
LOX	Development Hypoxia	Increased levels	Breast	Lung	Human/Mouse	[46]
LOXL2	EMT Hypoxia	Increased levels	Breast	Lung	Mouse	[51]
Zeb1	EMT Inflammation	Increased levels	Breast	Lung	Mouse	[52]
POSTN	Bone regeneration Cell adhesion	Increased levels	Breast/Cancer stem cells	Lung	Human/Mouse	[54]
VCAM-1	Cell adhesion	Increased levels	Breast	Lung/Bone	Mouse/Human	[56,90]
MMP-9	Metabolic processes	Increased levels	Breast	Brain	Mouse	[57]
PRAME	Apoptosis Differentiation	Increased levels	Melanoma	Brain	Human/mouse	[60]
VEGF-A	Angiogenesis	Increased levels	Melanoma Lung	Brain	Mouse	[66]
TSP-1	ECM constituent	Elevated levels	Breast	Bone Marrow	Mouse Zebrafish Human	[75]
CXCL12/CXCR4	Embryonic development	Elevated levels	Prostate	Bone marrow	Human/Ex vivo	[76]
CXCL5/CXCR2	Proliferation	Elevated levels	Breast	Bone	Mouse	[78]
E-selectin	Cell adhesion	Elevated levels	Breast	Bone marrow	Human/Mouse	[79]
PTHRP	Metabolic processes	Elevated levels	Breast	Bone	Mouse	[83]

**Table 2 ijms-20-06158-t002:** Therapeutic approaches targeting dormancy.

Approach	Mechanism	Therapeutic Method	Effect	Ref
**Prolonging Dormant State**				
Enhancing tumor-associated microvessel induced dormancy	Regulation of the IFNγ/IDO1/TSP1 axis	Administration of TSP1	Reduction of proliferation of invasive cells	[93]
Inhibiting angiogenesis	Activation of Angiostatin-regulated pathways	Upregulation of Angiostatin	Inhibition of tumor growth, reduction of metastases	[94]
Regulation of expression of LPA1 that is inversely correlated with Nm23-H1 expression	Modulating LPA1 levels	Specific LPA1 inhibitor Debio-0719	Reduced expression of proliferative markers Ki67 and pErk, increase of p-p38 stress kinase	[97]
Epigenetic regulation of expression of pluripotent genes	Upregulation of master receptor NR2F1	5-AZA demethylating agent	Increase in expression of SOX9, RARβ, and NANOG	[41]
Tumor blood vessel antigens	Activation of T-cell dependent immunity	Vaccines against TBVA	Tumor regression	[104]
Combination of immune therapy with Adriamycin	Activation of T cells and NK cells against cancer cells	Tumor-sensitized T cells and CD25(+) NKT cells	Sensitization of dormant cells to immunoediting, prolonged animal survival	[106]
**Elimination of Dormant Cells**				
Inhibition of cell cycle	Prevention of COL1-induced proliferation and upregulation of p27	Saracatinib with ERK1/2 inhibitor	Apoptotic cell death	[95]
Activation of T cells against cancer	Induction of NK cells to express PDL-1	Vaccination with cells transduced with CXCL10	Destruction of cancer cells by immune system	[105]
Reduce inflammation at metastatic site	Reducing the pro-metastatic effect of GM-CSF and IL-5	Low-fat diet	Decreased metastatic burden	[108]
Reduction of systemic inflammation	Inflammation	Perioperative treatment with NSAIDs	Decreased metastatic burden	[109]
Induction of mitochondrial dysfunction	Reduced mitochondrial respiration, leading to bioenergetic catastrophe	Small molecule VLX600	Tumor cell death	[110]
Reducing resistance induced by the JAK/STAT pathways	Inhibition of SOCS1 and IL-3	Specific inhibitors against SOCS1 and IL-3	Apoptosis	[111]
**Sensitization of Dormant Cells to Chemotherapy**				
Blocking communication of cancer cells with microenvironment	Disrupting CXCL12/ CXCR4 binding	CXCR4 antagonists	DTCs are mobilized from the BM, activate cell cycle	[92]
Blocking interaction with microenvironment	miRNA contents of exosomes	Administration of MSC loaded with antagomiR222/223	Breast cancer cells become sensitive to carboplatin	[96]
Cancer stem cells	Modulating Fra-1 levels	Enhanced expression of Fra-1	Decreased tumor incidence, chemosensitivity	[98]

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
