# Peer review of "Mechanisms of Metastatic Tumor Dormancy and Implications for Cancer Therapy"

_ijms, 2019, doi:10.3390/ijms20246158_

Round 1

Reviewer 1 Report

Comments to the Authors:
In their manuscript, Neophytouand Colleagues summarize the recent knowledge about the mechanisms of metastatic tumor dormancy and discuss about the development of novel therapeutic strategies to combat metastatic cancer by targetingthe dormancy stage. This review paper focuses on interesting issues in cancer research. However, the current study includes the following pointsthat need to be addressed.

Major issues:

Introduction section: the first paragraph (Lines 27-31) needs appropriate references.

Host immune system is required to maintain tumor cells in a state of functional dormancy (Schreiber R., et al. Science 331 (6024), 1565-1570.). The authors should describe and discuss this point in appropriate section of this review paper.

Making tables to summarize studies in the following sections in this reviewwould be greatly helpful to improve this review paper;

2). Intrinsic mechanisms regulating metastatic dormancy

3). Microenvironment

4). Therapeutic approaches against dormancy

Author Response

Reviewer 1

We would like to thank the reviewer for the constructive comments which have been fully addressed, as described below.

Major issues:

Introduction section: the first paragraph (Lines 27-31) needs appropriate references.

Response:

References 1 and 2 have been added to this section.

Host immune system is required to maintain tumor cells in a state of functional dormancy (Schreiber R., et al. Science 331 (6024), 1565-1570.). The authors should describe and discuss this point in appropriate section of this review paper.

Response:

 We have added a paragraph discussing the role of adaptive immunity in regulating cancer cell dormancy at metastatic sites under section 4, highlighted in green.

Making tables to summarize studies in the following sections in this review would be greatly helpful to improve this review paper;

2). Intrinsic mechanisms regulating metastatic dormancy

3). Microenvironment

4). Therapeutic approaches against dormancy

Response:

To address this, we have created Table 1 summarizing the intrinsic and microenvironmental mechanisms regulating metastatic dormancy and Table 2 summarizing the therapeutic approaches against dormancy.

Reviewer 2 Report

This is a comprehensive review on tumor metastatic dormancy. Although this topic is a complicated topic, authors have nicely summarized a recent advance in this field.  I would suggest following points.

Although this manuscript is predominantly focused on factors/signaling pathways involved in tumor-intrinsic dormancy, extrinsic dormancy should be (at least briefly) described in an independent section. The extrinsic dormancy is includes immune-mediated dormancy and angiogenic dormancy. In a recent paper (Blood. 2019 Jul 4;134(1):30-43), Khoo et al. nicely demonstrated the dormancy signature in a multiple myeloma model, and identified AXL as a key gene. This should be included in AXL part. Figure 1 requires major changes. In Figure 1, all reported factors that are up- or down-regulated in dormancy and escape conditions are described in several text boxes (such as FBWX7, cMYC, TGFR…). This could be misleading, given that dormant cells (or escaped cells) do not uniformly exhibit these features (i.e., inter-tumoral heterogeneity and intra-tumoral heterogeneity should exist in dormancy signature). Instead on listing all specific candidate pathways, it would be better to describe more broadly (i.e. impact of metabolic changes, hypoxia, and inflammation on cellular quiescence/reactivation). In the therapeutic section, it might be interesting to discuss how we can improve detection of dormant cancer cells clinically, given that selected patients with dormant tumor cells will require such therapeutic intervention to improve outcomes.

Author Response

Reviewer 2:

We would like to thank the reviewer for the constructive comments which have been fully addressed, as described below.

Although this manuscript is predominantly focused on factors/signaling pathways involved in tumor-intrinsic dormancy, extrinsic dormancy should be (at least briefly) described in an independent section. The extrinsic dormancy includes immune-mediated dormancy and angiogenic dormancy.

Response:

To address this, we have added a paragraph discussing the role of adaptive immunity in regulating cancer cell dormancy at metastatic sites under section 4, highlighted in green. We have also elaborated on the importance of angiogenic dormancy under section 3.3, highlighted in green.

In a recent paper (Blood. 2019 Jul 4;134(1):30-43), Khoo et al. nicely demonstrated the dormancy signature in a multiple myeloma model and identified AXL as a key gene. This should be included in AXL part.

Response:

We have added this paper discussing the key role of AXL in regulating myeloma cell dormancy in the bone, as a continuation of the AXL part (section 3.4.1), highlighted in green.

Figure 1 requires major changes. In Figure 1, all reported factors that are up- or down-regulated in dormancy and escape conditions are described in several text boxes (such as FBWX7, cMYC, TGFR…). This could be misleading, given that dormant cells (or escaped cells) do not uniformly exhibit these features (i.e., inter-tumoral heterogeneity and intra-tumoral heterogeneity should exist in dormancy signature). Instead on listing all specific candidate pathways, it would be better to describe more broadly (i.e. impact of metabolic changes, hypoxia, and inflammation on cellular quiescence/reactivation).

Response:

To address this issue, we have significantly improved Figure 1 as follows: To avoid loss of significant information for the readers we have decided to retain the list of reported genes/factors implicated in sustaining dormancy or promoting organ-specific escape from dormancy. To address the phenomenon of hererogeneity and more accurately present this information, we have now categorized this list of factors based on the impact of specific processes, such as hypoxia, inflammation, angiogenesis etc, on their expression levels during dormancy or escape stages.

In the therapeutic section, it might be interesting to discuss how we can improve detection of dormant cancer cells clinically, given that selected patients with dormant tumor cells will require such therapeutic intervention to improve outcomes. 

Response:

To address this important topic, in the revised manuscript we discuss detection methods of dormant cancer cells in the “Future Perspectives” section at the end of the Manuscript (Chapter 5).

Reviewer 3 Report

It is a very well written and comprehensive review of mechanisms governing tumor dormancy and tumor proliferation. 

Below a few suggestions for further improving the quality of the manuscript

1) Data from animal studies should be clearly separated from human data in all subsections

2) Authors give a large body of data regarding the mechanisms of tumor dormancy. However there is only one chapter devoted to therapeutic approaches. Authors should present more data on therapeutic methods 

3) Regarding therapeutic approaches, Authors should present in more detail data from clinical trials if any

4) It will be very interesting to give information about ongoing clinical trials

5) A section devoted to future perspectives should be included in the  manuscript 

Author Response

Reviewer 3:

We would like to thank the reviewer for the constructive comments which have been fully addressed, as described below.

1) Data from animal studies should be clearly separated from human data in all subsections.

Response:

We have now included this information in the newly formed Table 1.

2) Authors give a large body of data regarding the mechanisms of tumor dormancy. However there is only one chapter devoted to therapeutic approaches. Authors should present more data on therapeutic methods 

Response:

We have now added information regarding therapeutic methods which are presented in Table 2.

3) Regarding therapeutic approaches, Authors should present in more detail data from clinical trials if any:

4) It will be very interesting to give information about ongoing clinical trials

Response:

We have addressed both points 3 and 4 raised by the reviewer, by adding a paragraph discussing clinical trials at the end of section 4, highlighted in green.

5) A section devoted to future perspectives should be included in the manuscript.

Response:

This has been added as section 5 in the manuscript, highlighted in green.

Round 2

Reviewer 1 Report

The authors have addressed the reviewers' questions sufficiently.

Author Response

We would like to thank the reviewer for the constructive comments which have been fully addressed.